

# The object-specific flood damage database HOWAS21

Patric Kellermann[1], Kai Schröter[1], Annegret H. Thieken[2], Sören-Nils Haubrock[3], Heidi Kreibich[1]

[1] GFZ German Research Centre for Geosciences, Section Hydrology, Telegrafenberg, 14473 Potsdam, Germany

[2] Institute of Environmental Science and Geography, University of Potsdam, Karl-Liebknecht-Straße 24-25, 14476 Potsdam, Germany

[3] Beyond Concepts GmbH, Adolfstr. 23, 49078 Osnabrück, Germany

*Correspondence to*: Patric Kellermann (patric.kellermann@gfz-potsdam.de)

**Abstract.** The Flood Damage Database HOWAS21 contains object-specific flood damage data resulting from fluvial, pluvial and groundwater flooding in Germany. The datasets incorporate various pieces of information about flood impacts, exposure, vulnerability, and direct tangible damage at properties from several economic sectors. The main purpose of development and design of HOWAS21 is to support forensic flood analysis and the derivation of flood damage estimation models. This paper highlights exemplary analyses to demonstrate the use of HOWAS21 flood damage data in these two application areas. The data applications indicate a large potential of the database for fostering a better understanding and estimation of the consequences of flooding. HOWAS21 recently enlarged its scope and is now also open for international flood damage data.

## 1 Introduction

Natural hazard damage data are an essential component for ex-post event analysis and disaster response as well as for risk assessment and management (Hübl et al., 2002). A transparent and standardized data collection procedure is required to ensure comparability of such data (Merz et al., 2004). De Groeve et al. (2014) identify four main application areas for damage data, i.e. damage compensation, damage accounting, forensic analyses, and disaster risk modelling. Damage compensation refers to the use of damage databases for compensation mechanisms. For example, the European Union Solidarity Fund (EUSF) requires damage data for the substantiation of claims. Damage accounting is an application area aiming at the documentation of damage trends as well as the evaluation of disaster risk reduction policies, and often done on the national or international level using e.g. the global Emergency Events Database (EM-DAT) (www.emdat.be; Chorýnski et al., 2012; see also Sect. 2 and supplementary material). Damage data are also widely used for forensic analyses. Such analyses are performed e.g. to improve disaster management via quantification of the relative contribution of damage drivers such as exposure, vulnerability and coping capacity to the overall damage. The fourth main application area is natural hazards and risk modelling, which uses damage data to derive damage models for the estimation of disaster impacts. In general, the information needs for the four main application areas are overlapping, whereby, however, the forensic analyses and the disaster risk modelling applications require a higher level of detail. For example, the development, calibration and validation of flood damage models requires detailed, object-specific damage data as well as comprehensive information about exposure and vulnerability characteristics.



Hereby, when speaking of "object-specific", the damage data is recorded at the asset level, i.e. for individual affected elements such as e.g. a residential building or commercial property.

Flood damage is usually classified into direct and indirect damage. Direct flood damage occurs when exposed objects (or humans) have physical contact with flood water, whereas indirect flood damage is induced by direct damage and may also
occur outside of the flood event with respect to space and/or time (Merz et al., 2004). Further, both damage types can be distinguished into tangible damage, i.e. damage that can be adequately monetized, and intangible damage (Smith and Ward, 1998).

Already in the 1970s Grigg and Helweg (1975) identified water depth and object type (e.g. residential building) as the most important damage-influencing factors for flooding in urban areas. Consequently, depth-damage-curves describing the relation
of water depth and building damage differentiated according to building use became the internationally accepted standard approach for flood damage estimations in the built environment (Smith, 1994).

However, reported water depth and resulting object-specific damage usually show large scatter, and, thus, it is obvious that flood damage is influenced by more drivers besides water depth and building use (Blong 2004; Merz et al., 2004). Flood damage is influenced by factors such as flow velocity, duration of inundation, contamination of floodwater, availability and
information content of flood warning, and the quality of external response in a flood situation (Smith 1994; Wind et al., 1999; Penning-Rowsell and Green, 2000; Kreibich et al., 2005, 2009; Thieken et al., 2005; Totschnig and Fuchs, 2013). Also, Thieken et al. (2005) and Merz et al. (2013) investigated single and joint effects of impact (i.e. flood characteristics) and resistance factors (e.g. building type) on flood damage ratios in the residential sector. However, forensic knowledge on flood damage processes is scarce and there are still general shortcomings in empirical damage assessment on the object level. For
example, many studies of flood damage only include detailed assessments of a relatively small number of objects (Blong, 2004; Mazzorana et al. 2014). This is mainly due to the fact that reliable and detailed flood damage data, i.e. providing information about the affected object, flood warning, precautionary measures, etc., are still only scarcely available (e.g. Merz et al., 2010, Meyer et al., 2013, Wagenaar et al., 2016; Gerl et al., 2016; Menoni et al., 2016). Therefore, the effects of such drivers are mostly unknown and cannot be considered satisfactorily in damage modelling. Studies such as Ramirez et al.
(1988), Joy (1993), Gissing and Blang (2004), Thieken et al. (2005), Zhai et al. (2005) or Kreibich et al. (2007) particularly aimed at overcoming this lack of detailed damage data. Therefore, comprehensive information about flood damage, event and object characteristics, flood precaution variables, social factors and other damage-influencing factors of flood-affected residential and commercial properties was collected. Outside the scope of science, however, flood damage data are either not or only insufficiently collected, since damage data collection in the aftermath of a natural event is not mandatory and collection
standards hardly exist (Menoni et al., 2016, Thieken et al., 2016), or they are not made available by the data compilers (e.g. insurance industry) mainly for reasons of data security and property rights.

The main objective of the flood damage database HOWAS21 is to overcome these shortcomings. It was established at the German Research Centre for Geosciences (GFZ) in 2007 and incorporates object-specific flood damage data for a variety of economic sectors resulting from fluvial, pluvial and groundwater flooding (Thieken et al., 2009). HOWAS21 particularly aims



to support 1) forensic flood damage analyses, and 2) the derivation of flood damage models. This paper builds on the book chapter of Kreibich et al. (2017) and focuses on an expanded exemplification of possible analyses in these two application fields using the HOWAS21 data as a basis. After giving an overview of prominent flood damage databases (Sect. 2), a general description of the database concept and structure, the technical design, and the data sources of HOWAS21 is given in Sect. 3. Subsequently, descriptive statistics of available flood damage data in HOWAS21 are shown and a variety of exemplary

analyses in the application areas of forensic flood damage analyses and damage model derivation are performed and discussed with regard to usefulness of the object-specific flood damage data (Sect. 4). Finally, an outlook and recommendations are given (Sect. 5).

## 2 Overview of prominent flood damage databases

Depending on the requirements of the application area both the scale (granularity) of recording natural hazard damage data

and the scope (spatial coverage) of damage databases vary significantly (de Groeve et al., 2014). Many of the existing natural hazard damage databases, which also include flood damage, are event-specific databases. Such databases usually contain damage costs aggregated to the national level. This type of damage data provide a suitable basis for ex-post analysis of disastrous events (Hübl et al., 2002) as well as for damage accounting, e.g. trend analyses investigating whether damage due to natural hazards increases over time (Bouwer, 2011). Prominent examples of global event-specific damage databases are the

NatCatSERVICE database of Munich Re (www.munichre.com; Kron et al., 2012) and the EM-DAT. The NatCatSERVICE database contains overall and insured damage figures and fatalities of natural catastrophes around the globe. The natural events are classified in geophysical (e.g. earthquake), meteorological (e.g. windstorm), hydrological (e.g. flood, mudflow), and climatological events (e.g. extreme temperature, drought, wildfire). The EM-DAT comprises global data on the occurrence and impact of natural (floods, droughts, storms, mass movements, etc.), technological (e.g. industrial or transport accidents)

and complex disasters (e.g. famine) from 1900 to the present. An example for a continent-wide database is the European database HANZE, which contains information on dates, locations, and damage of 1564 flash, river, coastal, and compound floods between 1870 and 2016 (Paprotny et al. 2018). The HANZE database is open access via https://data.4tu.nl/repository/collection:HANZE. A well-known example of a national event-specific database is the Swiss flood and landslide damage database established at the Swiss Federal research Institute WSL (Hilker et al., 2009). This database

contains systematically documented information on floods and mass movements (i.e. direct monetary damage) as well as injured people and fatalities in Switzerland since 1972 using press articles as the main source of information. Broader overviews of event-specific databases are provided by e.g. Tschoegl et al. (2006), Gall et al. (2009), De Groeve et al. (2014) or Rudari et al. (2017). In recent years, many initiatives were launched at national and European Union (EU) levels to improve the availability and usefulness of damage data. For example, an ongoing EU initiative aims at the standardization of damage

databases. Based on a defined conceptual framework, this initiative provides technical recommendations for the development





of EU guidelines for recording disaster impacts with the central aim of translating the Sendai Framework for Disaster Risk Reduction into action (Corbane et al., 2015).

Forensic flood damage analyses as well as damage model derivation and validation predominantly require object-specific data which permit in-depth investigations of causal relations between hazard, exposure, vulnerability, and damage magnitudes (e.g.

Downton et al., 2005; Jonkman, 2005). However, object-specific databases are still rare. The Flood Hazard Research Centre (FHRC) from Middlesex University, United Kingdom (UK), maintains a (national) object-specific flood damage database comprising mainly synthetic damage data generated via expert estimations about expected damage in case of a certain flood intensity (what-if-analyses). This synthetic data is complemented, whenever possible, by empirical data sourced from e.g. on-site surveys or insurance companies. The corresponding absolute flood damage functions for the UK are published in the

Multi-Coloured Manual (Penning-Rowsell et al., 2013), as well as in its predecessors (Penning-Rowsell and Chatterton, 1977; Parker et al., 1987). The Austrian Federal Railways (ÖBB) holds an object-specific flood damage database for railway infrastructure and operation in Austria (Moran et al., 2010; Kellermann et al., 2016). This database incorporates information about the affected infrastructure object and resulting service disruptions, the structural damage and corresponding repair costs, the hazard characteristics, and possible mitigation measures. The ÖBB Natural Hazard Management uses the detailed

information internally as a basis for the development and implementation of both structural and non-structural risk reduction measures (Kellermann et al., 2016). In Italy, a new national database was developed and recently integrated into the Italian Civil Protection system. The main objective of this database is the development of specific depth-damage curves for Italian contexts. For this, new procedures for data collection and storage were developed and applied at the local level for the residential and commercial sectors (Molinari et al., 2014). An extended overview of existing flood damage databases, including

general characteristics and references for further information, is provided in the supplementary material of this paper (Table S1). The examples given therein do not claim to be exhaustive but serve to illustrate the variety of global efforts to collect (and learn from) natural hazards damage data.

The predecessor database of HOWAS21 was the HOWAS database, which was developed and maintained by the German "Bund/Länder-Arbeitsgemeinschaft Wasser" (LAWA) (see e.g. Buck and Merkel, 1999; Merz et al., 2004). HOWAS

contained professionally surveyed damage information of approx. 4000 properties affected by nine flood events between 1978 and 1994 in Germany (see also Sect. 3.3). Only direct tangible flood damage to buildings was considered for HOWAS, distinguishing between damage to the building fabric, the fixed inventory, and the movable inventory. Similar to the concept of HOWAS21 (see Sect. 3.1), the data was classified into six damage sectors, i.e. private sector, public infrastructure (e.g. fire station), service sector (e.g. restaurant), mining and building industry (e.g. carpentry), manufacturing (e.g. beverage industry),

and buildings for agriculture, forestry and horticulture (Merz et al., 2004). The damage records represented repair costs (building damage) or replacement costs (inventory damage), and all costs were given in absolute values. The object-specific flood damage data is further complemented by additional information about damage-influencing factors (e.g. floor space) whenever possible. Due to missing cost coverage, LAWA stopped the database maintenance in 1994.



## 3 HOWAS21 database characteristics

### 3.1 Concept and structure

HOWAS21 is a relational database hosted and administrated by the GFZ, which is also responsible for compiling, reviewing, and maintaining consistency of data, assigning access rights and verifying user requests. HOWAS21 does not benefit from funding for data collection or updating and, thus, is largely relying on voluntary data contributions from e.g. surveys and data acquisition campaigns. The use of HOWAS21 follows a community-based concept and is organized in three user groups with
different levels of data usage (Thieken et al., 2009): The World user group is designed for the interested public and grants access to a range of general information and evaluations. Data selection options comprise structured queries, filtered by catchment area, regions (provinces and municipalities), periods (event year), sectors, data acquisition methods, and combinations thereof. Additionally, the HOWAS21 interface offers two registered user groups. Registered user group I is reserved for any kind of institution that provides a defined amount of data to HOWAS21. Full access to the entire database is
granted to this user group. Users from academia or non-commercial projects, who do not provide data, can apply for the registered user group II. Those users are granted permission for a restricted project-specific use of the database. In return, a feedback on project results based on HOWAS21 data is requested. Moreover, in case that flood damage data is collected at a later stage, these shall be provided to HOWAS21. The scope of use, the reporting requirements, and the prohibition of data dissemination are regulated via a user contract. 61 users from science, insurance, authorities, and engineering consultancy
registered to HOWAS21. Among these, 12 organizations provided data to HOWAS21 (user group I). This ratio indicates that, as yet, the majority of organizational users are mainly interested in extracting data, but hardly willing or able to contribute data to the database.

The data structure for HOWAS21 was derived from a multi-step online expert survey based on the Delphi-approach. The central idea of this approach is to reach a consensus among the respondents by having a questionnaire filled several times,
after receiving feedback of earlier responses of all participants. Complementing the HOWAS21 database, a manual outlining the theoretical framework for flood damage assessment and a suggestion for damage documentation was developed (Thieken et al. 2009).

The HOWAS21 database classifies object-specific damage into six damage sectors:

1. private households
2. commercial and industrial sector, including public municipal infrastructure (administration, social affairs, education, etc.) as well as agricultural buildings
3. agricultural and forested land
4. public thoroughfare, including roads and transport infrastructure
5. watercourses and hydraulic structures (particularly flood defense structures)
6. urban open spaces



The attributes of individual damage cases are grouped into three (partly sector-specific) database tables as shown exemplarily in Table 1. Such attributes include information about the flood characteristics at the location of the affected object (i.e. hazard), object characteristics (i.e. vulnerability), and the extent of damage and damage mitigation (i.e. consequences). Moreover, additional meta-information is provided for each damage case, including information about the flood (e.g. event year,

catchment name, seasonality, flood type) and the data acquisition campaign (e.g. survey type, period of the survey, sample description). Minimum data provision requirements for damage cases to be incorporated into HOWAS2l were defined as follows:

- economic sector of the affected object
- monetary damage
- inundation depth
- year (month) of the event
- spatial location of the affected object at least on the level of zip codes or municipalities

These requirements are set based on the rationale of ensuring the possibility to link flood damage to hydraulic impact, whereby water depth was found the most important explanatory variable for flood damage in a variety of studies (e.g. Merz et al., 2013;

Vogel et al., 2018).

In total, HOWAS21 incorporates a broad range of hazard variables (e.g. flow velocity, flood duration, and contamination), vulnerability parameters such as building characteristics (e.g. building shape, year of construction) and precautionary measures (e.g. warning time, type and effectiveness of measures), and flood consequences (e.g. absolute and relative damage of flood-affected objects, economic damage due to business interruption in the commercial sector.

The HOWAS21 concept further includes a procedure to determine the general quality of flood damage data. The approach is based on the hierarchical framework of Wang and Strong (1996) and assesses the quality via scores ranging from 0 (poor quality) to 4 (very good quality). More detailed information and examples of the data quality assessment concept applied in HOWAS21 can be found in Kreibich et al. (2017).

HOWAS21 data are in an anonymous format respecting personal rights according to data privacy regulations.

**3.2 Technical design**

HOWAS21 consists of two major components: the flood damage database and a web application. The database contains and manages access to all available flood damage records. The HOWAS21 web application complements the database by providing a user-friendly, internet-based data access interface. This interface is directly accessible using a standard web browser following the URL http://howas21.gfz-potsdam.de and can be used to visualize, analyze, and download HOWAS21 data. Users

are provided query functionality in the database on selectable criteria, such as catchments, damage sectors, or year of event.





The technical design of HOWAS21 aims to take account of the complexity and heterogeneity of flood damage data in multiple sectors. Corresponding metadata is comprehensive and allows for putting the damage information into context. A variety of attributes (e.g. river catchment, flood event year, damage sector) can be used to filter and analyze the data.

Users can register for a HOWAS21 registered user group (see Sect. 3.1) via the web application. If access is granted and a data
usage contract is signed, registered users can access the database, analyze, and download data.

### 3.3 Data sources

The Flood Damage Database HOWAS21 is designed for empirical flood damage data. A significant part of the data origins from the predecessor database of HOWAS21 - the HOWAS database (see Sect. 2). It was developed and maintained by the German Working Group on water issues of the Federal States and the Federal Government (LAWA) from 1978 to 1994 (Buck
and Merkel, 1999; Merz et al., 2004). Damage data of HOWAS were collected via on-site expert surveys by damage surveyors of insurance companies and used as a basis for financial compensation, wherefore these damage estimates are considered to be reliable.

Further, an essential portion of HOWAS21 damage data result from computer-aided telephone interviews (CATI) with private households and companies who suffered flood damage in the years 2002, 2005, 2006, 2010, 2011 and/or 2013 (e.g. Kienzler
et al., 2015; Thieken et al., 2016). Potential participants of CATI were identified by compiling lists of affected streets with the help of e.g. flood masks derived from radar satellite data, or publicly available information such as official reports and press releases (e.g. Kreibich et al., 2007; Thieken et al., 2007; Kreibich et al., 2011). The interviews were mainly carried out by pollsters and, for every interview, it was consistently sought to consult the person with the best knowledge about the flood event as well as the resulting object-specific damage.

For the damage sectors "public thoroughfare" and "watercourses and hydraulic structures", only a few damage datasets are implemented in HOWAS21 so far. All these datasets are collected via on-site expert inspection after the 2002 food in Dresden. Damage data for public thoroughfare comprises 246 inundated sections of road infrastructure in the City of Dresden affected during the Elbe river flood in 2002. The dataset includes physical road characteristics (e.g. length, width, sidewalks), the road classifications, and additional object features. With respect to damage the data was collected in two steps: First, the absolute
monetary damage was derived from reports of the city administration providing the reconstruction costs of affected road sections, whereby no unit length is defined for the road sections. Second, the magnitude of structural damage was quantified on a six-point scale, and the condition of the road before the flood was quantified on a five-point scale by experts from the city administration (Kreibich et al., 2009). A similar procedure was applied for the 525 damage cases along watercourses and hydraulic structures in the City of Dresden also affected in the course of the 2002 Elbe flood.



## 4 Exemplary analyses

HOWAS21 aims at compiling comprehensive flood damage data (i.e. including object-specific hazard, exposure, and vulnerability characteristics) to support forensic flood damage analyses as well as flood damage model derivation. In this section, available HOWAS21 flood damage data are characterized and used for exemplary analyses to demonstrate the potential of the database in both of these application areas.

The up-to-dateness of the data used for the analyses is November 01, 2019. Damage values distinguish between building damage, contents damage, and damage to goods and stock, whereby damage to goods and stock is only defined for the commercial and industrial sector. Further, damage in the sectors public thoroughfare and water courses and hydraulic structures is by definition classified as building damage. All costs are given in Euros and the reference year of an individual cost value is the year of the related flood event occurrence. Thus, in order to achieve comparability, all costs were converted to the year 2018. Conversion factors are the price indexes for construction works on residential buildings and the consumer price indexes for replacement costs of household contents as well as commercial and industrial goods and stock, both published by the Federal Statistical Office Germany.

### 4.1 General descriptive statistics

HOWAS21 comprises a total number of 8558 object-specific flood damage records from flood events between 1978 and 2013 in Germany. The geographical distribution of the damage data is depicted in Fig. 1. The private households sector accounts for the by far largest data fraction (57.1%), followed by the commercial and industrial sector (33.9%). The sectors water courses and hydraulic structures (6.1%), and public thoroughfare (2.9%) are as yet rather underrepresented (see Table 2). No data are yet available for the sectors agricultural and forested land, and urban open spaces. In fact, the commercial and industrial sector does contain a small number of flood affected agricultural buildings.

Both in respect to the number of damage records and the level of detail of information, i.e. the number of different hazard, exposure, and vulnerability variables, HOWAS21 is the most comprehensive flood damage database for empirical data worldwide. Although the current set of damage records provides data for a number of variables exceeding the minimum requirements of HOWAS21 (see Sect. 3.1), most of the records are far from exhaustively filling the defined sector-specific variable space. For example, for any damage record in the private households sector, there is currently a 42% chance of data availability for non-mandatory variables. This chance even decreases to around 22% for the commercial and industrial sector. In turn, for certain other non-mandatory variables such as building type or building shape (in the variable space for the private households sector) the data availability is close to 100%. Generally, the availability of detailed flood damage data is often limited due to the facts that damage data collection in the aftermath of a flood is not mandatory, sufficient and properly trained personnel is mostly not available, and collection standards do not exist (Menoni et al., 2016; Thieken et al., 2016).

The HOWAS21 flood damage data features a total mean damage of EUR 238,366 for the commercial and industrial sector, which is roughly six times the mean damage for the private households sector (EUR 39,994) (price level in 2018). Damage





data in the sectors public thoroughfare and watercourses and hydraulic structures are yet only available for the City of Dresden during the 2002 Elbe flood.

The histograms of total damage per sector are displayed in Fig. 2. Due to their positive skewness, the data samples are plotted
on the logarithmic scale. The data shows large variability, which is demonstrated graphically by the histogram width, and numerically by the coefficients of variation (CoV) showing values between 187% (public thoroughfare) and 616% (commercial and industrial sector). The variability probably stems largely from the heterogeneity of the (as yet) incorporated flood years in terms of flood characteristics, affected exposure, and vulnerability characteristics of this exposure. In particular, as indicated by the exceptionally high CoV, the commercial and industrial sector is characterized by strong heterogeneity,
thereby hampering the flood damage estimation in this sector (see also Sect. 4.3).

### 4.2 Gaining insights into flood damage processes

Forensic analysis techniques attract growing interest in science and risk management since they help to uncover the complex underlying causes and effects of disasters (Wenzel et al., 2013; Dolan et al., 2017). More specifically, such analyses are performed e.g. to quantify the relative contribution of damage drivers such as exposure, vulnerability and coping capacity to
the overall damage. Other applications include the assessment of interdependencies of damage drivers, and the change of the correlation between a specific damage driver (e.g. water depth) and the resulting consequence (damage) over time (e.g. between different events in the same region).

In this section three examples of forensic flood damage analyses on the basis of HOWAS21 flood damage data for the private households sector are given, namely 1) the examination of interactions between flood damage-influencing variables by means
of a Principal Component Analysis (PCA) (Sect. 4.2.1), 2) the estimation of variable importance for multivariate flood damage modeling using Random Forest (Sect. 4.2.2), and 3) the investigation of relative changes of the relation between water depth and absolute flood damage to buildings over time (i.e. between individual flood events) based on multilevel linear regression (Sect. 4.2.3). All forensic analyses are based solely on building damage in the private households sector caused by fluvial floods, i.e. damage cases attributed to pluvial as well as groundwater flooding were removed from the data samples, to facilitate
the interpretation of flood damage processes.

### 4.2.1 How do potential flood damage-influencing variables interact?

An important task for a better understanding of flood damage processes is the investigation of potential damage influencing variables and their interactions. For this, the correlation structure of such variables and their contribution to the total variance in the HOWAS21 data (related to building damage in the private households sector) is examined by means of a PCA. A
Principal Component (PC) is a normalized (z-transformation) linear combination of the original variables capturing the maximum variance in a dataset. Hence, each PC explains a certain percentage of the total variance in the dataset, whereby the first PC by definition explains the largest share of total variance, the second PC explains the second largest share, and so forth.





Moreover, all PCs are uncorrelated (i.e. perpendicular) to each other and, altogether, they reflect the underlying structure in the given dataset.

All damage influencing variables for which non-categorical data are (currently) available in HOWAS21 were considered for the PCA, and variables quantifying flood consequences (e.g. absolute building damage in Euros) were neglected. The data sample was furthermore centered and scaled to avoid bias in the variance of the data due to variable scale mismatches and, consequently, false estimations of the PC.

According to the Kaiser criterion as well as the scree plot, four significant PCs can be identified in the HOWAS21 data sample.
They explain around 59.9% of the total variance, whereby approx. one third of this variance can be attributed to the first PC (21.4% of total variance). In order to facilitate the interpretation of the variable contributions to each PC and, thus, to gain insights into the interaction of variables a varimax rotation was applied. Table 3 provides the loadings of the eleven potential flood damage-influencing variables on the four significant PCs. Loadings with absolute values equal or above 0.5 are considered to be high and, hence, are given priority in the interpretation of the respective PCs. It appears that the first PC is
dominated by the hazard variable water depth, whereby, although not exceeding the threshold of 0.5, the variables flood duration and number of floors also show a relatively high loading. The second PC is marked by a mixture of hazard and damage mitigation variables (see also Table 1), namely the variables flow velocity class and lead time. The two remaining PCs are largely characterized by the exposure characteristics year of building construction (PC3), and building type (PC4). Although also not exceeding the defined loading threshold of 0.5, the variables equipment class and building area show a relatively
notable loading on PC3 or respectively PC4.

In order to supplement the investigation of variable interactions with an estimation of the influence of the PCs on flood damage, the correlation between the factor scores of each PC and the absolute building damage was analysed (Table 3). Results show that absolute building damage correlates best with PC1, i.e. the component driven by the flood impact variable water depth. Lower (but still statistically significant) correlation is further given to PC2, PC3, and PC4.

The PCA shows that flood impacts variables, particularly water depth, are the factors with highest influence on absolute building damage. These are, however, closely followed by a variety of exposure and vulnerability characteristics, in particular lead time, year of building construction, and building type. When looking at the explained variance of only around 59.9% for the extracted PCs, the results further indicate that a large number of variables is needed to sufficiently explain the total variance in the HOWAS21 data for fluvial flood damage to private households. This supports the results of e.g. Schröter et al. (2014)
showing that flood damage processes are intrinsically complex and, thus, can be better described using a variety of explanatory variables representing different flood damage processes. Consequently, the findings basically underline the importance of a comprehensive damage data collection approach as followed by HOWAS21.

**4.2.2 How important are individual variables for multivariate flood damage estimations?**

Building on the insights into variable interactions in the HOWAS21 data (see Sect. 4.2.1), a logical next step towards a better
understanding of flood damage processes is to investigate the individual variable importance for multivariate damage





modeling. To do so, the Random Forest variable importance algorithm was used to identify non-monotonic and multivariate relationships of the damage-influencing variables (i.e. the predictors) to estimate absolute building damage in the private households sector. However, in contrast to the variable selection used for the PCA (see Sect. 4.2.1) and due to the applicability in a Random Forest framework, also categorical variables (i.e. roof type and building shape) are included in this analysis.

The variable importance of a predictor is estimated by random permutation of values of this particular variable. The idea is that this random permutation leads to an increase in the prediction error compared with the error generated by original values. Accordingly, the mean increase in prediction error caused by permutation of a certain variable serves as a measure for the importance of this particular variable.

Figure 3 shows the results of the Random Forest variable importance analysis. Using the Mean Square Error (MSE) as the

decisive prediction error statistic, the variable water depth is ranked as the most important variable for damage estimation, closely followed by the variables lead time and flood duration. These three predictors show an increase in the MSE between approx. 8% and 12%. Exposure characteristics such as building area and year of construction also play a notable role according to the MSE increase of around 4-5%. On the other hand, the variables flow velocity class, number of floors and roof type show only very low importance (see Fig. 3).

Overall, similar to what could be inferred from the PCA with regard to variable interactions and variance in the data (see Sect. 4.2.1), a variety of different hazard, exposure, and vulnerability characteristics are relevant information also for multivariate flood damage modeling in the private households sector. The variable lead time again appears to play a significant role also in multivariate flood damage modeling. Vogel et al. (2018) came to a similar conclusion when identifying lead time as an important predictor for flood damage estimations using a Bayesian Network. Generally, the results of a variable importance

analysis can also help to obtain a better understanding of the (correctness of the) model`s logic as well as to improve the model by e.g. removing unimportant variables (see Sect. 4.3).

### 4.2.3 Has the relation of flood impact and resulting damage changed over the years?

Another important step towards a more thorough understanding of flood damage processes is seen in the examination of the process dynamics. For example, by comparing the functional relationship between flood damage drivers and the resulting

damage for different years of flooding, potential patterns or trends over time could be identified. Against this background, the following exemplary analysis is aimed at identifying potential trends in the linear relation between water depth and absolute building damage to private households between different flood years using multilevel regression.

Multilevel models, also known as e.g. mixed models, hierarchical models or group-effects models, are useful for analyses involving hierarchical, nested, clustered, or longitudinal data. Hierarchical data, for example, consist of units (e.g. flood-

affected objects) which can be grouped into other units (e.g. flood events), whereby the grouped units represent a distinct data level (i.e. level 1), and the grouping unit forms a superior data level in the hierarchy (i.e. level 2). Multilevel models allow e.g. to assess the amount of data variability due to each data level and, thus, to explicitly identify and investigate group effects. Thus, these model features are useful also for forensic flood damage analyses and, in particular, for investigating changes of





relations between a predictor variable (e.g. water depth) and the response variable (e.g. absolute building damage) from one
flood event to another (i.e. the group effect in our case). More detailed information about principles of multilevel models can
be found in e.g. Gelman and Hill (2006).

In a first step, the relevance of group effects in the HOWAS21 data sample is measured by means of the Intra-class Correlation
Coefficient (ICC). The ICC quantifies the ratio of variance on the hierarchical data level 1, i.e. the level of flood-affected
objects, being explained by data level 2, i.e. the flood events. For example, an ICC value of 0.2 indicates that 20% of the total
data variance lies between the groups, and 80% within the groups accordingly. Common practice suggests to consider ICC
values of 0.05 or higher as an indicator for significant group effects (LeBreton and Senter, 2008). In such cases multilevel
models should be favored over simple linear models. Since the ICC value of the damage data of absolute building damage to
private households amount to approx. 0.08, a multilevel linear regression model is applied to further investigate the expected
changes in the depth-damage relation between individual flood events.

The scatter plot of water depth and damage to building structure including multilevel linear regression lines is shown in Fig.
4. Looking at the lines reveals a more or less continuous increase of both the intercept and the slope with increasing flood
event year. This trends can be further investigated when plotting the group effects of the model (see Fig. 4). Three main
findings emerge: First, the high group variability of both regression parameters among the 13 flood years clearly confirms
significant group effects as already suggested by the ICC. Second, the range of group-specific residuals indicate that, on the
whole, flood damage data of years of flooding further back in time tend to scatter more compared with more recent events in
the 21st century (see Fig. 4). Third, the development of the group effects of depth-damage relations over time points to an
overall increasing trend in the flood damage (see also Fig. 3). In other words, the majority of regression models derived from
flood events, which occurred in the 21st century, show higher intercepts as well as higher slopes relative to the values of a
simple linear regression model, whereas flood events of the 20th century mostly have lower parameter values, respectively.
For example, comparing the group effects of the flood year 2013 with the flood year 1988, the mean damage in 2013 is nearly
EUR 26k higher than the mean damage as derived from the entire HOWAS21 building damage data, whereas the mean damage
in 1988 is almost EUR 9k lower (price level in 2018). Further, an increase of one cm of water depth leads to an increase of
absolute building damage being around EUR 103 higher (year 2013) than the mean increase, or approx. EUR 35 lower (year
1988), respectively.
Summing up, based on water depth as the determinant and absolute building damage as the response variable, multilevel
regression results reveal considerable flood damage process dynamics between individual flood years, which is manifested by
changes in both the intercept and the slope of regression lines from one flood year to another (see Fig. 4 and Fig. 5). These
changes or, more specifically, the overall positive trend in the occurred flood damage with time, can have manifold reasons
such as i) the increase in event severity in terms of depth and/or area, ii) the increase in exposure in terms of number of objects
and/or asset values, iii) the increase in vulnerability of affected objects, and iv) changes of data collection methods. More
(forensic) analyses would be required to better attribute the observed trend to potential causes. For example, the current
multilevel model is based on the entire HOWAS21 dataset for the private households sector, i.e. involving a region-unspecific



series of flood years. In order to reduce the effects of changes of exposure characteristics and, thus, to focus more on hazard and vulnerability-related influences, the input data for regression could be limited to a series of flood events that occurred in
the same region (e.g. river catchment or Federal State). More sophisticated approaches to assess spatio-temporal variability in flood damage processes are presented in Sairam et al. (2019).

## 4.3 Model derivation and validation

The following section presents examples of different flood damage models that can be derived on the basis of HOWAS21 data, and briefly evaluates their performance. These include 1) a variety of univariate, i.e. linear, polynomial, and square-root,
depth-damage curves for all four economic sectors as well as all damage types available in HOWAS21 (i.e. damage to buildings, damage to contents, damage to goods and stock), and 2) a multi-variate Random Forest regression model for absolute damage to building structures in the private households sector.

Univariate depth-damage curves as well as underlying data are plotted in Fig. 6. It appears that for most of the combinations of economic sector and damage type the different regression types result in similar depth-damage curve progressions. An
exception is, however, the public thoroughfare sector, for which the polynomial regression curve is noticeably undulating. Generally, the visual evaluation of the curves fitted to the considerably scattered data samples already indicates that univariate depth-damage relations, i.e. the use of water depth as a sole predictor, only partly (and often insufficiently) explain the complexity of flood damage processes.

The suitability of the models to estimate absolute flood damage was evaluated by means of three different error measures: the
Mean Bias error (MBE), the Mean Absolute Error (MAE), and the Root Mean Square Error (RMSE). These are calculated based on one third of the respective original data sample as validation, whereby the remaining two thirds were used for the model derivation.

The error statistics for all models are summarized in Table 4. Generally, according to RMSE and MAE, the three univariate regression types perform similarly in estimating absolute flood damage for all damage types. Based on the MBE, the model
performances in some cases differ significantly, whereby, however, none of the models consistently performs best (or worst) across all sectors and damage types. Also, all univariate models except the models for damage to contents in the private households sector show a MAE being in a comparable order of magnitude as the observed mean damage of the sector (see Fig. 2 and Table 4).

When used for regression, Random Forests are ensembles of a (large) number of regression trees. Each regression tree is
constructed by recursive binary splits of a bootstrap sample from the original data called bagging, and each binary split can be related to any predictor variable at any value. The data not included in the bootstrap sample to train a regression tree, the so-called Out-Of-Bag observations (OOB), are used for the calculation of model performance measures as well as estimations of predictor variable importance (see Sect. 4.2.2). A more detailed description of the Random Forest algorithm can be found e.g. in Breiman (2001). Random Forests are generally seen useful for flood damage modeling, since they are applicable to both





categorical and continuous data, they allow for non-linear and non-monotonous input data, and they are able to capture
       predictor interactions (e.g. Merz et al., 2013; Schröter et al., 2014; Kreibich et al., 2016; Sieg et al., 2017; Sultana et al., 2018).
       Due to the low importance of the predictor variables flow velocity class, number of floors and roof type (see Sect. 4.2.2, Fig.
       3), they are excluded from the Random Forest model derivation. Consequently, the Random Forest model is learned on the
       basis of eight predictor variables, of which two variables (water depth and flood duration) are hazard characteristics, five
variables (building type, building shape, building area, year of construction, equipment class) represent exposure
       characteristics, and one variable (lead time) addresses damage mitigation.

       In comparison with the univariate modeling approaches, the multivariate Random Forest model shows better results, although
       the estimation errors are still very high when viewed in relation to the observed mean damage (see Fig. 2 and Table 4). Overall,
       the findings of this model derivation and validation exercise suggest a limited capacity of univariate models to explain the
complex flood damage processes. The error measures indicate large estimation uncertainties for all univariate models, whereby
       the regression type (i.e. linear, square-root, and polynomial) has only marginal influence on the individual model performance
       irrespective of the economic sector. The largest errors are observed in the commercial and industrial sector, which can be
       explained by the strong heterogeneity of this sector (see Fig. 2, Sieg et al., 2017). Already in 1999, using damage data from
       the predecessor database HOWAS, the Department of Water Resources Management and Rural Engineering of the University
of Karlsruhe (IWK) showed that the derivation of generally valid damage functions is difficult, in particular for damage
       categories being poorly represented. The lack of comprehensiveness of flood damage data also led Merz et al. (2004) to the
       conclusions that the HOWAS database is not totally representative for flood damage in Germany, and that the use of empirical
       flood damage data involves considerable uncertainties. They therefore claimed that i) flood damage data should be collected
       at the object level whenever possible to better support the development and validation of flood damage models, and ii) the data
base should be enlarged with regard to the number of variables to follow a more comprehensive and systematic data collection
       approach. Both of these claims were also directly addressed in the successor database HOWAS21. Indeed, adding new
       predictors significantly improves the flood damage estimates as can be clearly seen from the example of the Random Forest
       performance in the private households sector (see Table 4) – whereby, although the Random Forest estimates are still subject
       to considerable uncertainties. Again, this result is in line with Schröter et al. (2014) showing that complex models better capture
the multidimensional nature of flood damage processes.

**5 Conclusions**

       The Flood Damage Database HOWAS21 incorporates object-specific data about flood impact, exposure, vulnerability, and
       direct tangible damage in various economic sectors resulting from fluvial, pluvial and groundwater flooding in Germany. Its
       strengths include data quality features, the compliance with strict minimum requirements for data entries, the integration of
sectors such as public thoroughfare or watercourses being widely underrepresented in damage data collections, and the
       consideration of a multitude of damage-influencing variables.
These features are also essential and integral components of the HOWAS21 concept to support forensic flood damage analyses as well as the development of flood damage models. The exemplary analyses presented in this paper give a hint on the large potential of this database for such application fields. They further confirm two central findings of other relevant studies, i.e.

the fundamental role of the hazard variable water depth to estimate flood damage on the one hand, and the need for a variety of different explanatory variables to better understand and describe the intrinsically complex flood damage processes on the other hand.

The HOWAS21 database has been further developed and optimized for the hosting of object-specific flood damage data with a global scope. This extension of scope of HOWAS21 includes, inter alia, the incorporation of a globally valid spatial identifier

and the international standard classification for economic activities.

However, it depends on the cooperation and commitment of the (scientific) community to provide flood damage data to HOWAS21, so that the empirical flood damage database continuously growths and as such increases its value for the whole community. The higher the amount of data and the more diverse the contained data is (e.g. from different flood types and regions, covering various sectors), the better it can support forensic flood damage analyses as well as the development of flood

damage models. Therefore, if flood damage data is or becomes available, we expressly encourage data owners to include it into HOWAS21 for their own benefit (i.e. getting access to all data contained in the database) but even more important for the benefit of the whole community. Check out HOWSAS21 via http://howas21.gfz-potsdam.de/howas21/ or e-mail howas21@gfz-potsdam.de.

**Supplementary Material**

Table S1

**Author Contributions**

Conceptualization, P.K., K.S., A.H.T. and H.K.; Formal analysis, P.K.; Writing—original draft, P.K., K.S., A.H.T., S.-N.H. and H.K.

**Conflict of Interests**

The authors declare that they have no conflict of interest.

**Acknowledgements**

The authors gratefully acknowledge all existing data providers for their valuable contribution to HOWAS 21, namely: Deutsches GeoForschungsZentrum GFZ, Universität Potsdam, Deutsche Rückversicherung AG, LAWA Bund/Länder-



Arbeitsgemeinschaft Wasser, alpS - Zentrum für Naturgefahren und Risikomanagement GmbH, Deutsches Institut für
Wirtschaftsforschung e.V. (DIW Berlin), Hydrotec Ingenieurgesellschaft für Wasser und Umwelt mbH, JBA Risk
Management, Landeshauptstadt Dresden, Leuphana Universität Lüneburg INFU - Institut für Umweltkommunikation, Riocom
Ingenieurbüro für Kulturtechnik und Wasserwirtschaft, Risk Management Solutions Inc., Swiss Reinsurance Company Ltd.

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



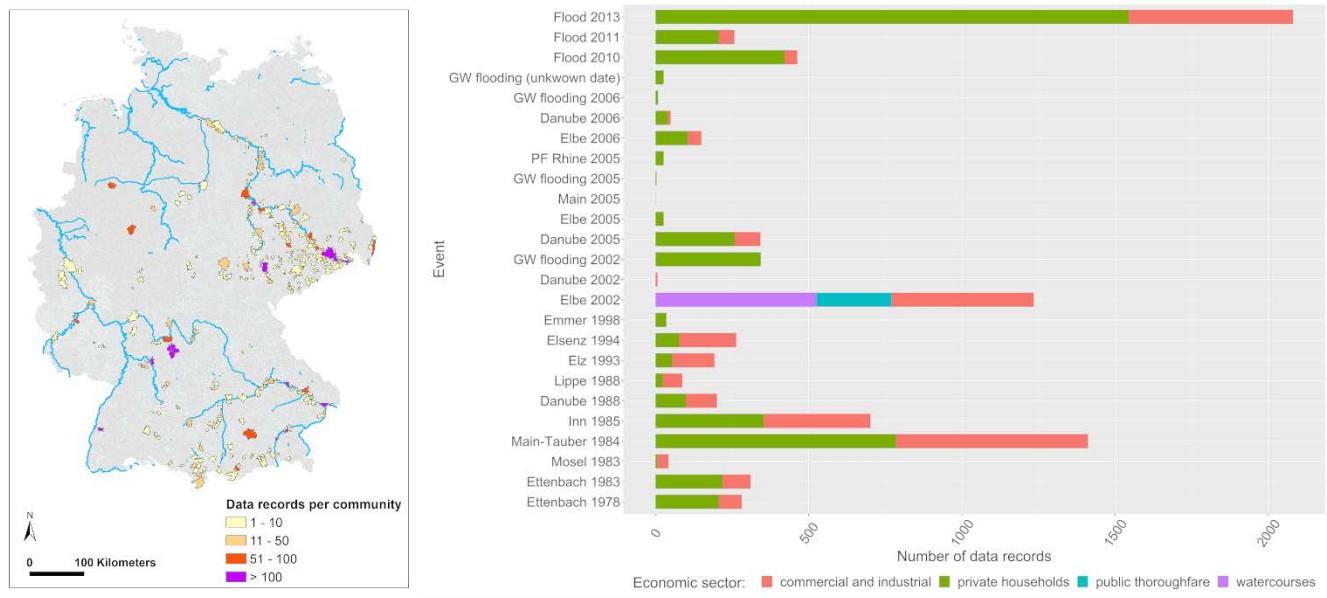

**Figure 1: Left plot: Locations of available flood damage data in HOWAS21 (adapted from Kreibich et al., 2017); Right plot: Number of data records per economic sector and event (GW = groundwater; PF = pluvial flood; own illustration). No data are yet available for the sectors agricultural and forested land and urban open spaces.**

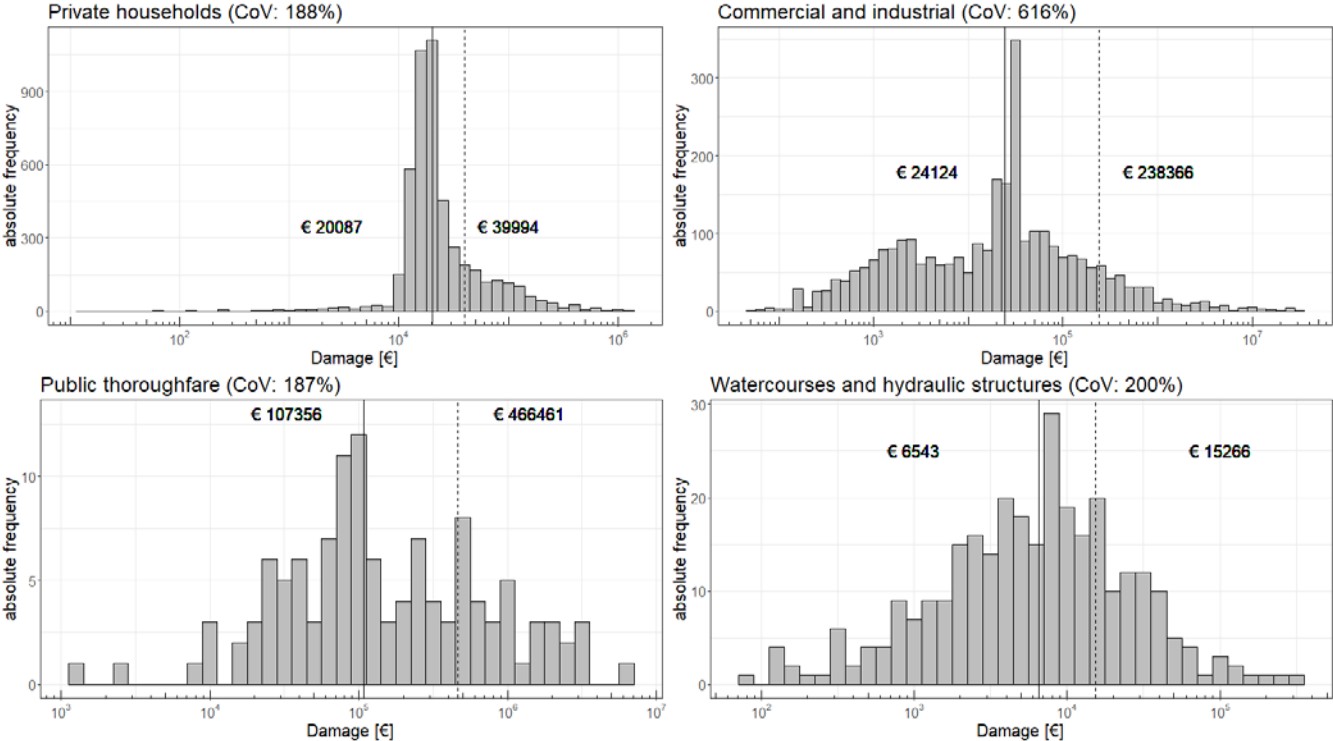

**Figure 2: Histograms of total damage values per sector for all records of the HOWAS21 database. Data are plotted on the logarithmic scale. Solid and dashed lines indicate the median and the mean values, respectively.**

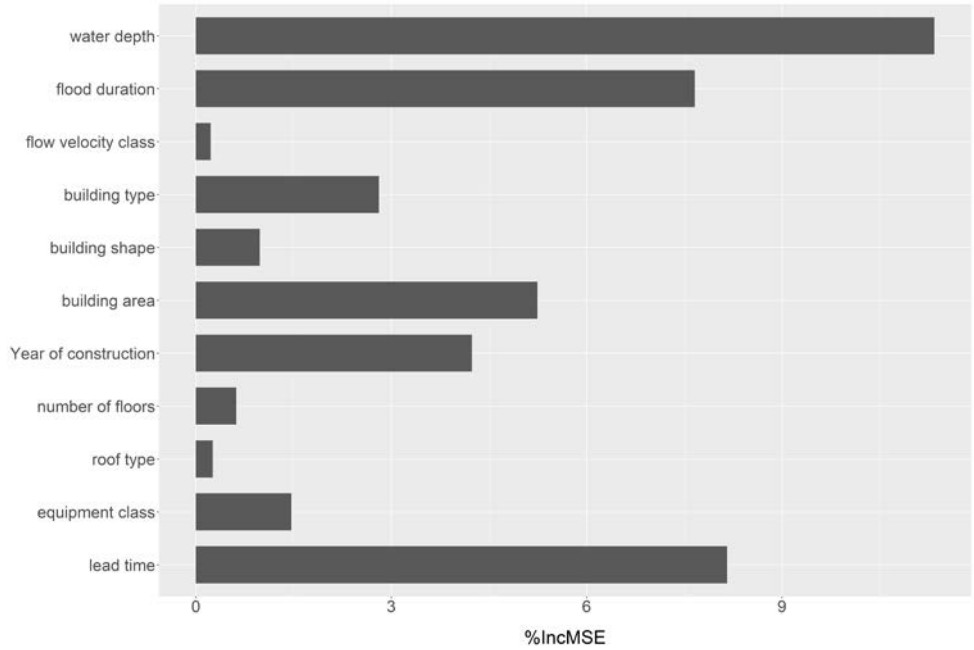

**Figure 3: Random Forest variable importance for the estimation of building damage in the private households sector (n = 1610).**
**Importance measure is the Mean Square Error (%IncMSE).**


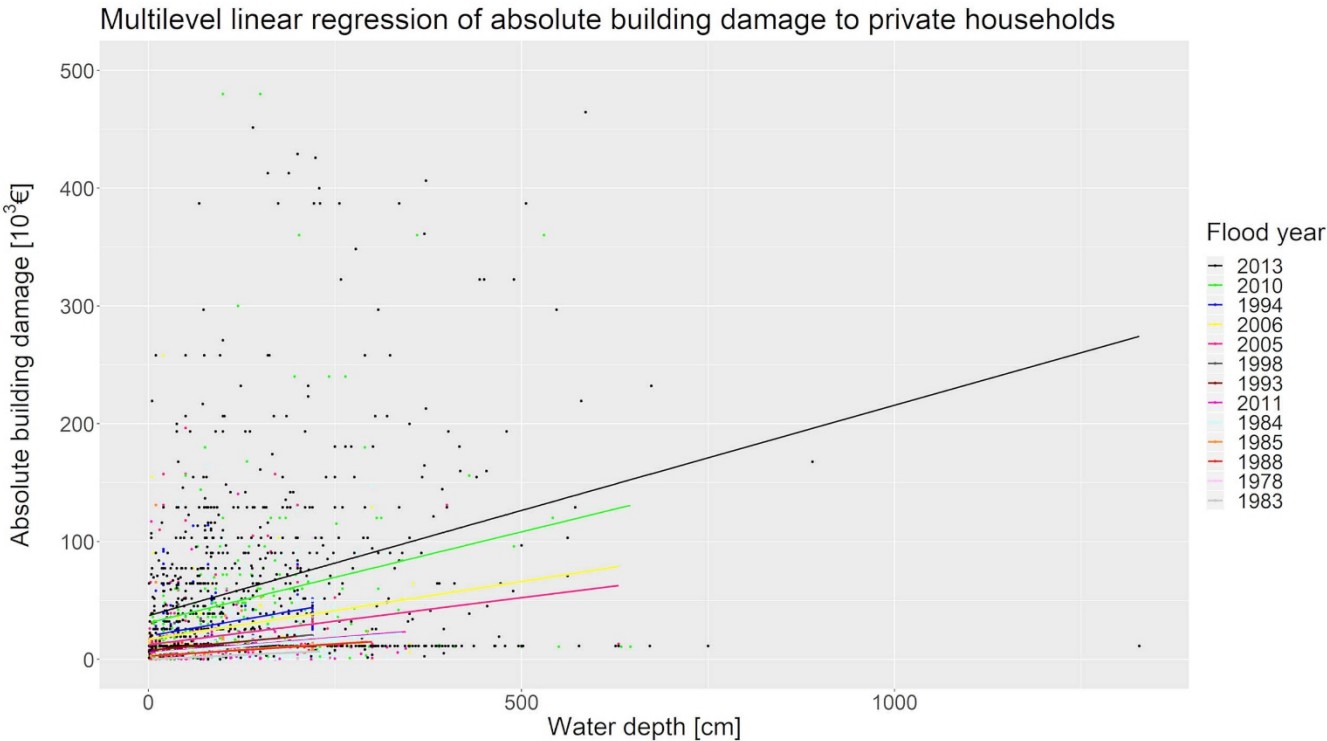

**Figure 4: Scatter plot of water depth and absolute damage to buildings in the private households sector including multilevel linear regression lines. Regressions are carried out on the basis of the following samples sizes: n=207 (1978), n=226 (1983), n=785 (1984), n=354 (1985), n=124 (1988), n=55 (1993), n=78 (1994), n=36 (1998), n=286 (2005), n=146 (2006), n=243 (2010), n=90 (2011), n=984 (2013).**





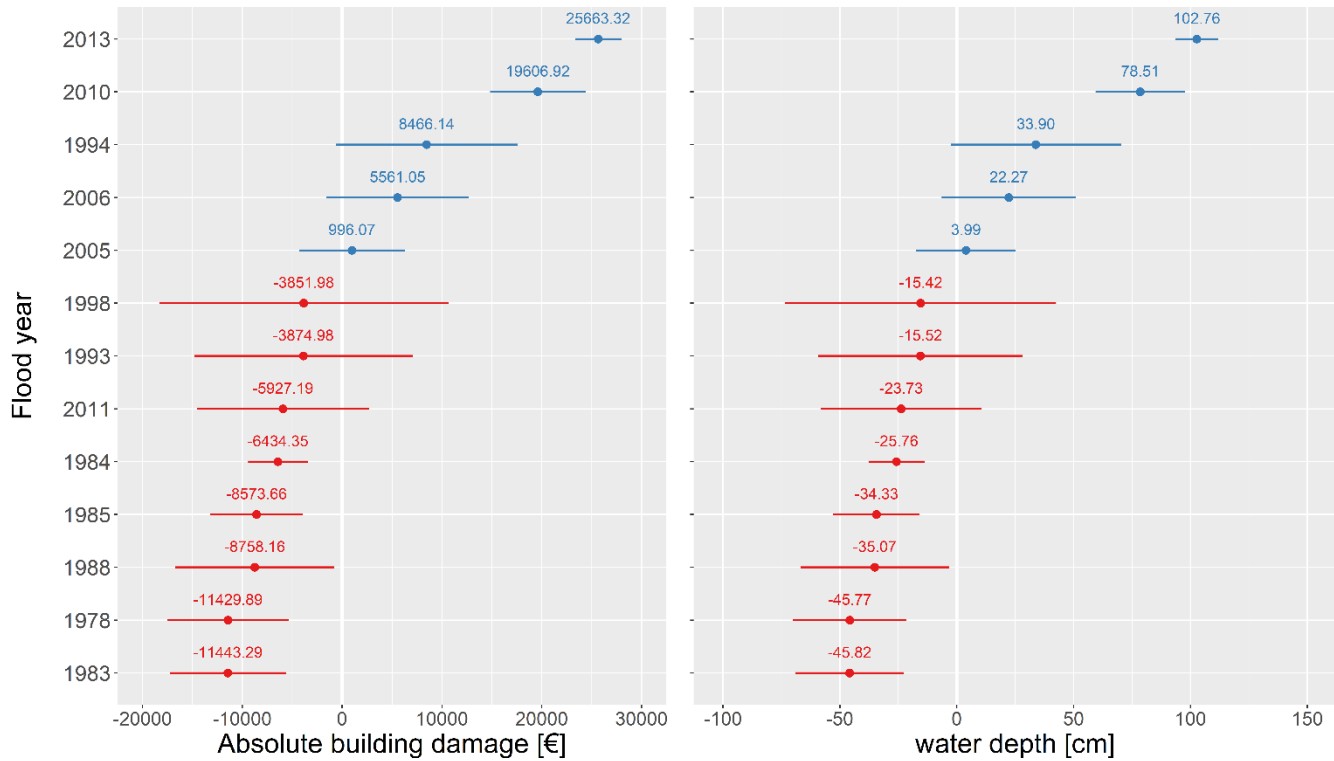

**Figure 5: Dot-and-Whisker plot of group effects of the multilevel linear regression model for absolute damage to buildings in the private households sector. The left plot corresponds to the random intercept, i.e. the variation of the intercept per group from the overall intercept of a simple linear regression. Each group represents a particular flood year. The right plot depicts the random slopes. The plotted values are the mean differences between the intercept for each group and the overall intercept (or slope for each group and the overall slope). The whiskers correspond to the range of group-specific residual values.**






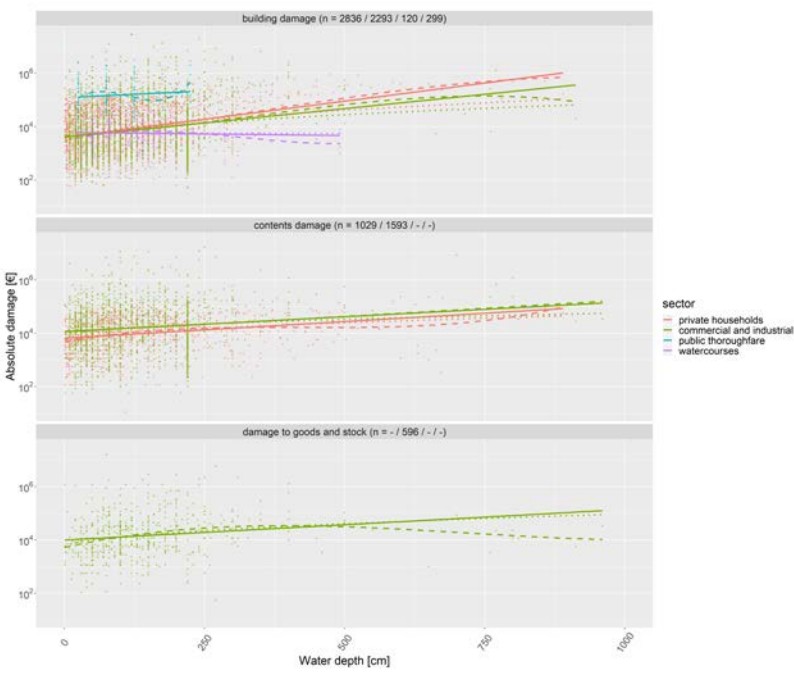

**Figure 6: Depth-damage curves for all four economic sectors (private households, commercial and industrial, public thoroughfare, watercourses) and damage types (building damage, contents damage, damage to goods and stock) currently available in HOWAS21. Solid line = linear regression; dashed line = polynomial regression; dotted line = square-root regression; Absolute damage values are plotted on the logarithmic scale. The numbers in brackets in the plot titles are the sample sizes of the datasets, whereby the order 670 of numbers corresponds to the order of elements in the plot legend.**





**Table 1: Exemplary overview of the main damage information tables for private households (adapted from Thieken et al., 2009, and Kreibich et al., 2017).**

| Flood characteristics at the location of the affected object[1] | Object characteristics and damage information[2] | Damage mitigation[3] |
|---|---|---|
| <ul><li>Start, end, duration of inundation at the object</li><li>Name of river causing the inundation</li><li>Maximum water depth</li><li>Maximum flow velocity</li><li>Contamination, flotsam</li><li>Local return period</li><li>Hazard peculiarities, description of hazard at object</li></ul> | <ul><li>Location of the building</li><li>Building type and characteristics (number of stories, age, quality, net dwelling area, intrusion paths and intake sill, building material, use of the cellar, etc.)</li><li>Value of the building, building damage, damage ratio</li><li>Contents value, contents damage, damage ratio</li></ul> | <ul><li>Knowledge about the flood hazard</li><li>Precautionary measures</li><li>Early warning, lead time</li><li>Emergency measures</li><li>Effectiveness of measures</li></ul> |

[1] Items are used for all sectors.


[2] Specific items per sector.

[3] Items for private households and commercial/industrial sector only.



**Table 2: Total number of damage records per economic sector in HOWAS21 and average data availability rate for non-mandatory variables according to the defined minimum requirements for data incorporation into HOWAS21.**

| Economic sector | Total number of damage records | Fraction of total number (%) | Average data availability rate for non-mandatory variables (%) |
|---|---|---|---|
| Private households | 4882 | 57.1 | 42 |
| Commercial and industrial | 2905 | 33.9 | 21.8 |
| Public thoroughfare, road and transport infrastructure | 246 | 2.9 | 51.7 |
| Watercourses and hydraulic structure | 525 | 6.1 | 43.5 |
| Agricultural and forested land | 0 | - | - |
| Urban open spaces | 0 | - | - |
| Sum | 8558 | 100 | - |






**Table 3: Loadings of variables potentially influencing flood damage to buildings in the private households sector[1].**

| Variables | Principal Components (n=1610)[2] | | | |
| --- | --- | --- | --- | --- |
| | PC1 | PC2 | PC3 | PC4 |
| Water depth | **-0.60** | -0.36 | 0.21 | 0.12 |
| Flood duration | 0.46 | 0.20 | 0.19 | -0.29 |
| Flow velocity class | -0.28 | **0.63** | -0.28 | -0.06 |
| Building type | -0.09 | 0.12 | -0.01 | **0.70** |
| Building area | -0.35 | -0.18 | 0.21 | -0.49 |
| Year of building construction | 0.13 | 0.05 | **0.73** | -0.03 |
| Number of floors | 0.42 | --0.28 | 0.05 | 0.28 |
| Equipment class | -0.13 | 0.19 | 0.44 | 0.29 |
| Lead time | 0.08 | **-0.52** | -0.27 | 0.01 |
| | | | | |
| Pearson correlation coefficient (absolute building damage) [3] | **0.24** | **-0.19** | 0.05 | 0.04 |

[1] Principal Component analysis with varimax rotation; total variance explained is 59.89%.

[2] Significant Principal Components according to the Kaiser criterion. Bold numbers indicate absolute variable loadings $\geq 0.5$.

[3] Bold numbers indicate significant correlation coefficients based on a level of 0.01 (two-sided).



**Table 4: Performance of depth-damage curves and the Random Forest model as measured by error statistics (MBE: Mean Bias Error, MAE: Mean Absolute Error, RMSE: Root Mean Square Error). All values are given in EUR rounded to the nearest whole number.**

| | Private households – Building damage | | | | Private households – Contents damage | | | Commerce and industry – Building damage | | | Commerce and industry – Contents damage | | |
|---|---|---|---|---|---|---|---|---|---|---|---|---|---|
| Regression type | linear | polynomial | square-root | Random Forest | linear | polynomial | square-root | linear | polynomial | square-root | linear | polynomial | square-root |
| MBE | -495 | -1001 | -17 | 465 | -4310 | -4283 | -4286 | 69492 | 70072 | 70147 | 7673 | 9628 | 8389 |
| MAE | 37216 | 36531 | 38428 | 26160 | 22044 | 21945 | 21939 | 220387 | 221165 | 221091 | 213673 | 221184 | 216299 |
| RMSE | 67811 | 66656 | 69425 | 60171 | 33383 | 33310 | 33313 | 530451 | 531983 | 530206 | 760529 | 768589 | 762263 |

| | Commerce and industry – Damage to goods and stock | | | | Public thoroughfare – Building damage | | | Watercourses – Building damage | | |
|---|---|---|---|---|---|---|---|---|---|---|
| Regression type | linear | polynomial | square-root | - | linear | polynomial | square-root | linear | polynomial | square-root |
| MBE | 60162 | 58679 | 60198 | - | -130733 | -153774 | -133343 | 536 | -151 | 560 |
| MAE | 207454 | 208833 | 206790 | - | 403075 | 425304 | 405729 | 15443 | 16101 | 15436 |
| RMSE | 1108349 | 1110070 | 1108723 | - | 506249 | 516233 | 505145 | 30710 | 30941 | 30723 |
