# Peer review of "The object-specific flood damage database HOWAS 21"

_Natural Hazards and Earth System Sciences, 2019_

## Referee Comment (RC1) · Anonymous Referee #1 · 9 Mar 2020

Journal: NHESS
Title: **The object-specific flood damage database HOWAS21**
Author(s): Kellerman et al.
MS No.: NHESS-2019-420
MS Type: Research Article
**Iteration: First review**

The objective of the paper is to present the flood damage database HOWAS21, and to demonstrate its use to support forensic flood investigation and for the derivation of flood damage models. Accordingly, first the database is introduced, with respect to both conceptual and technical contents, and data sources. Then, exemplary analyses of the two application areas are supplied and commented.

The paper is generally well written and organised, methods and results are quite well explained; in this regard, I would suggest only minor revisions. Still, my doubt is about the suitability of the paper for publication in NHESS. In detail, two main aspects must be considered: 1) Most of the contents (at least from pg. 1 to pg. 7, out of 13 pages) replicate the book chapter of Kreibich et al. (2017) quoted in the paper; 2) the "new" part of the manuscript is simply limited to application of well-known analysis tools on HOWAS21 data, without supplying any additional knowledge to the state of the art, e.g. on flood damage explicative variables and their weight, on data interpretation tool (N.B., exceptions are represented by section 4.1 and 4.2.3 that however supply data of interests only for Germany). Of course, in the assumption that authors do not want to propose new models for the German context.

Given this premises, I suggest the rejection of the paper and supply some minor comments in the following.

**General comments**

Many repetitions are present (see minor comments); please, check and correct.

An unexplored aspect that may represent an additional value to the work is to show how information on data quality can be used for forensic investigation and damage modelling.

**Minor comments**

**Section 1: Introduction**

I found this section the most critical in terms of contents and structure. I suggest to re-organise the order of contents. In detail:

Pg. 1 line 17: "A transparent and standardized data collection procedure is required to ensure comparability of such data (Merz et al., 2004)". This sentence is not linked with previous and following ones. I suggest moving after the discussion ending at pg. 2 line 61

Pg. 2 line 33: "Flood damage is usually classified into direct and indirect damage. Direct flood damage occurs when exposed objects (or humans) have physical contact with flood water, whereas indirect flood damage is induced by direct damage and may also occur outside of the flood event with respect to space and/or time (Merz et al., 2004). Further, both damage types can be distinguished into tangible damage, i.e. damage that can be adequately monetized, and intangible damage (Smith and Ward, 1998)" → discussion of damage types is out of the scope of the manuscript and, however, it is not linked here with previous and following paragraphs. I suggest removing

Pg. 2 line 37 – 58: I found discussion on damage assessment approaches and damage influencing factors misleading at this point of the paper. I suggest to anticipate, before the discussion, the objective of HOWAS21 as supporting forensic analysis and damage model developing.

Moreover:

Pg. 1 line 24: "Such analyses are performed e.g. to improve disaster management via quantification of the relative contribution of damage drivers such as exposure, vulnerability and coping capacity to the overall damage" → I would add hazard among damage drivers

Pg. 1 line 29: "For example, the development, calibration and validation of flood damage models requires detailed, object-specific damage data as well as comprehensive information about exposure and vulnerability characteristics" → and hazard

**Section 2**

Pg. 3 line 85: "An example for a continent-wide database" → I would say "continent-wide event database"

Pg.3 line: "In recent years, many initiatives were launched at national and European Union (EU) levels to improve the availability and usefulness of damage data. For example, an ongoing EU initiative aims at the standardization of damage databases. Based on a defined conceptual framework, this initiative provides technical recommendations for the development, of EU guidelines for recording disaster impacts with the central aim of translating the Sendai Framework for Disaster Risk Reduction into action (Corbane et al., 2015)" → This sentence is not linked with previous and following ones. I suggest removing.

Pg. 4 line 11: "The main objective of this database is the development of specific depth-damage curves for Italian contexts. For this, new procedures for data collection and storage were developed and applied at the local level for the residential and commercial sectors (Molinari et al., 2014)" → As to my knowledge, this is not correct. The main objective of the database (i.e. FloodCAT) is to address knowledge needs required by the Floods Directive. There are not standardised procedures for data collection linked to the database, and the database works at the event level (not object level). Please, amend.

**Section 3**

Pg. 5 line 148: "The data structure for HOWAS21 was derived from a multi-step online expert survey based on the Delphi-approach. The central idea of this approach is to reach a consensus among the respondents by having a questionnaire filled several times, after receiving feedback of earlier responses of all participants" → I think an extended description of the adopted Delphi methodology is required for readers who are not familiar with previous works of the authors' team.

Pg. 6 line 161: "The attributes of individual damage cases are grouped into three (partly sector-specific) database tables as shown exemplarily in Table 1" → I understand Table 1 refers only to sector 1 & 2. If this is the case, please specify.

Pg. 6 line 176: "In total, HOWAS21 incorporates a broad range of hazard variables (e.g. flow velocity, flood duration, and contamination), vulnerability parameters such as building characteristics (e.g. building shape, year of construction) and precautionary measures (e.g. warning time, type and effectiveness of measures), and flood consequences (e.g. absolute and relative damage of flood-affected objects, economic damage due to business interruption in the commercial sector" → REPETITION, see pg. 6 line 162-163

Pg. 6 line 188: "This interface is directly accessible using a standard web browser following the URL http://howas21.gfz-potsdam.de and can be used to visualize, analyze, and download HOWAS21 data" → authors should specify that the interface is only in German, this can be a big issue for the use of the database for international data

Pg. 6 line 189: "Users are provided query functionality in the database on selectable criteria, such as catchments, damage sectors, or year of event" → REPETITION, see pg. 5 line 136

Pg. 7 line 192: "A variety of attributes (e.g. river catchment, flood event year, damage sector) can be used to filter and analyze the data" → REPETITION, see previous comment

Pg. 7 line 198: "It was developed and maintained by the German Working Group on water issues of the Federal States and the Federal Government (LAWA) from 1978 to 1994 (Buck and Merkel, 1999; Merz et al., 2004). Damage data of HOWAS were collected via on-site expert surveys by damage surveyors 200 of insurance companies and used as a basis for financial compensation, wherefore these damage estimates are considered to be reliable" → REPETITION, see section 2

Pg. 7 line 213: "…in the City of Dresden affected during the Elbe river flood in 2002" → REPETITION, see the line above and line 219

Pg. 7 line 216: "Second, the magnitude of structural damage was quantified on a six-point scale, and the condition of the road before the flood was quantified on a five-point scale by experts from the city administration (Kreibich et al., 2009)" → this is not clear, neither are the motivation of this process. Please, explain

**Section 4**

Pg. 8 line 221: "HOWAS21 aims at compiling comprehensive flood damage data (i.e. including object-specific hazard, exposure, and vulnerability characteristics) to support forensic flood damage analyses as well as flood damage model derivation" → REPETITION, said many times; please, consider removing

Pg. 8 line 228: "Further, damage in the sectors public thoroughfare and water courses and hydraulic structures is by definition classified as building damage" → why? not clear; please, specify

Pg. 8 line 234: "Both in respect to the number of damage records and the level of detail of information, i.e. the number of different hazard, exposure, and vulnerability variables, HOWAS21 is the most comprehensive flood damage database for empirical data worldwide" → Have authors evidence on this?

Pg. 8 line 246: "in the variable space for the private households sector" → What does it mean? Please, specify

Pg. 8 line 247: "Generally, the availability of detailed flood damage data is often limited due to the facts that damage data collection in the aftermath of a flood is not mandatory, sufficient and properly trained personnel is mostly not available, and collection standards do not exist" → this is partly true for HOWAS21 as data were all collected by trained personnel, also data from the original HOWAS. Isn't it?

Pg. 9 line 252: " Damage data in the sectors public thoroughfare and watercourses and hydraulic structures are yet only available for the City of Dresden during the 2002 Elbe flood" → REPETITION, see comment above

Pg. 9 line 263: "More specifically, such analyses are performed e.g. to quantify the relative contribution of damage drivers such as exposure, vulnerability and coping capacity to the overall damage" → I would replace with "More specifically, such analyses are performed e.g. *to understand* the relative contribution of damage drivers such as *hazard*, exposure, vulnerability and coping capacity to the overall damage"; as most of forensic investigations are conducted in the social/economic domain, on the bases of qualitative data.

Pg. 9 line 265: "Other applications include the assessment of interdependencies of damage drivers, and the change of the correlation between a specific damage driver (e.g. water depth) and the resulting

consequence (damage) over time (e.g. between different events in the same region)" → I would not say these are typical applications in forensic investigation.

Section 4.2.1 → it is not clear to me what authors mean here with "interactions among damage-influencing variables" and how the performed analysis gives information on that; please, explain

Pg. 11 line 341: "the following exemplary analysis is aimed at identifying potential trends in the linear relation between water depth and absolute building damage to private households between different flood years using multilevel regression" → how such information can support decision-makers? Please, specify

Pg. 12 line 360-374 → check figure numbers

Pg. 12 line 363: "the high group variability of both regression parameters among the 13 flood years clearly confirms significant group effects as already suggested by the ICC" → not clear, please specify

Pg. 13 line 394: "An exception is, however, the public thoroughfare sector, for which the polynomial regression curve is noticeably undulating" → Could authors explain why?

**Table and Figures**

Figure 5 → please check caption; I guess on the right slope is depicted

Figure 6 is not comprehensible: too little and too blurry

Table 1 → I would add a column for damage attributes

---

## Referee Comment (RC2) · Anonymous Referee #2 · 24 Mar 2020

The paper presents the flood damage database HOWAS21, and its use to support both forensic flood investigation and flood damage models. The topic is significant and has a great scientific interest. Nevertheless, I think that the paper should be organized in a more strict way, reducing the long descriptions, avoiding randomized examples and clearly dividing different features discussed using bullet points.

The parts of the paper describing the db are not very clear and the link with floods that caused damage is missing. Moreover, I suggest to describe the procedure more clearly because it is not clear how other researchers could easily implement the same kind of analysis.

In the following the points that in my opinion could be improved:

[Figure]

 c Section 2 is long and sometimes confusing. It should be rearranged. I think that, in order to help the reader to follow the description, table S1 should be included in the paper (formatted in a more concise way, by clustering groups of similar db). I also suggest to introduce some bullet points or sub-headings, because currently this section is long and dispersive.

 c Section 3.1: I suggest to schematize this part in order to be more effective. Firstly, the type of users should be clearly divided and not described altogether (also a diagram could be useful). At line 176: "In total, HOWAS21 incorporates a broad range of hazard variables" and after this the Authors presented examples randomized of these variables. I suggest to list in a table all the variables. This is a paper based on a db: it is impossible for the reader understand the discussion without a clear idea of the variables included. I understand that the db has been presented in previous papers, for this reason a simple and clear table can be easily prepared and can be more explicative than a series of examples.

 c Section 4.1. This section starts with damage described per type of damaged element without talk about the events that caused damage. How many flood events are included? What types are the most frequent? What the regions affected. This focus on the effects (damage) neglecting the causes (floods) also affect figure 1.

 c Fig. 1 is not homogeneous. On the Y axis I see "flood 2013; Elbe 2006; GW flooding 2006". What is the criterion to name the events analyzed? The basin? I don't think, because it is not reported in all the records. The year? I don't think, because there is one record without the year. The type of flood? I don't think, because it is not specified in all the cases. This is an important point to homogenize because if the events are well known to the authors, they can be unknown for readers. It is also unclear what the x axis reports. Maybe the number of cases of damage? "Number of data records" is very vague. In the map: what "community" means? Village? Town? Municipality? Prefecture?It is unclear what the colored areas represent.

[Figure]

• Concerning the variables reported in fig 3, it is not clear how some of them are measured. For example to me it is not clear how "building shape" or "roof type" are represented as numerical variables.

• Conclusions are not very representative of the contents of the paper, I suggest to review.

• Fig. 6 is absolutely not readable.

---

## Author Comment (AC1) · 29 May 2020

**Authors response to Referee #1**

*Dear referee,*
*we would like to thank you for the time and effort put into reviewing the manuscript. This response carefully addresses all the comments (our response is marked with an R: and written in italics). We further attach a change tracked version of the manuscript from which the changes proposed can be seen, see below this response.*
*Best regards*
*Heidi Kreibich on behalf of all co-authors*

Referee #1
The objective of the paper is to present the flood damage database HOWAS21, and to demonstrate its use to support forensic flood investigation and for the derivation of flood damage models. Accordingly, first the database is introduced, with respect to both conceptual and technical contents, and data sources. Then, exemplary analyses of the two application areas are supplied and commented.
The paper is generally well written and organised, methods and results are quite well explained; in this regard, I would suggest only minor revisions. Still, my doubt is about the suitability of the paper for publication in NHESS. In detail, two main aspects must be considered: 1) Most of the contents (at least from pg. 1 to pg. 7, out of 13 pages) replicate the book chapter of Kreibich et al. (2017) quoted in the paper; 2) the "new" part of the manuscript is simply limited to application of well-known analysis tools on HOWAS21 data, without supplying any additional knowledge to the state of the art, e.g. on flood damage explicative variables and their weight, on data interpretation tool (N.B., exceptions are represented by section 4.1 and 4.2.3 that however supply data of interests only for Germany). Of course, in the assumption that authors do not want to propose new models for the German context.
Given this premises, I suggest the rejection of the paper and supply some minor comments in the following.

*R: 1) We do not agree that the manuscript is replicating the book chapter. Meanwhile, the database HOWAS21 has been further developed technically and content wise: it is now available in German and English and is now also designed to integrate international flood damage data from other countries besides Germany. Since this improvement has just been finalized in May 2020, no international data has been integrated, yet. However, this can start soon. Still, we believe that this improvement of the database HOWAS21, which is newly described in the (revised) manuscript, is of high interest to the international readership of NHESS. However, we admit, that we have not stressed this sufficiently nor described these improvements in sufficient detail in our original manuscript. We will focus on this in the revised version.*
*2) We neither agree that the analyses we present are well-known and do not present new knowledge. Such detailed flood damage data as available in HOWAS21 is still scarce and so are multi-variable statistical analyses of flood damage data, which are necessary to better understand and model flood damage processes. In fact, this is the first time that old data from the 1980s and 1990s are analysed together with and are compared to data from the 2010s. We believe that the presentation of several exemplary analyses of the HOWAS21 data provides on the one hand motivation for the community to increase efforts for the collection of more such detailed object-specific damage data, undertake more such forensics analyses and develop better damage models beyond depth-damage functions. On the other hand, novel results about damage processes are presented, since this comprehensive dataset of more than 8000 damage cases has not been analysed before. The fact that the analyses are only based on data from Germany cannot call into question their scientific value. All data-based analyses about natural hazards rely on empirical data from case studies, just commonly on significantly smaller sample sizes, more localized areas, and often only covering one event. With a total number of 8558 object-specific flood damage records from several flood events between 1978 and 2013*

*affecting large areas in Germany, the HOWAS21 database is as such unique, and so are the results of the analyses.*

**General comments**

Many repetitions are present (see minor comments); please, check and correct.

*R: thanks for pointing this out, we removed unnecessary repetitions in the text.*

An unexplored aspect that may represent an additional value to the work is to show how information on data quality can be used for forensic investigation and damage modelling.

*R: In the context of forensic analysis and damage modelling, information about data quality could be used to highlight topics/findings that are uncertain and should be improved by further / more accurate data or better models. However, we believe that this is out of the scope of this paper.*

**Minor comments**

**Section 1: Introduction**

I found this section the most critical in terms of contents and structure. I suggest to re-organise the order of contents. In detail:

Pg. 1 line 17: "A transparent and standardized data collection procedure is required to ensure comparability of such data (Merz et al., 2004)". This sentence is not linked with previous and following ones. I suggest moving after the discussion ending at pg. 2 line 61

*R: We have improved the introduction, so that we hope it has now a clearer structure and reads smoother. We have deleted the mentioned sentence.*

Pg. 2 line 33: "Flood damage is usually classified into direct and indirect damage. Direct flood damage occurs when exposed objects (or humans) have physical contact with flood water, whereas indirect flood damage is induced by direct damage and may also occur outside of the flood event with respect to space and/or time (Merz et al., 2004). Further, both damage types can be distinguished into tangible damage, i.e. damage that can be adequately monetized, and intangible damage (Smith and Ward, 1998)" → discussion of damage types is out of the scope of the manuscript and, however, it is not linked here with previous and following paragraphs. I suggest removing

*R: Agreed, sentences are deleted.*

Pg. 2 line 37 – 58: I found discussion on damage assessment approaches and damage influencing factors misleading at this point of the paper. I suggest to anticipate, before the discussion, the objective of HOWAS21 as supporting forensic analysis and damage model developing.

*R: This part of the introduction is now closer linked to the main application areas for damage data, which are presented at the beginning of the introduction. We prefer to describe, as common, the objective of the paper and as such of HOWAS 21 in the last paragraph of the introduction.*

Moreover:

Pg. 1 line 24: "Such analyses are performed e.g. to improve disaster management via quantification of the relative contribution of damage drivers such as exposure, vulnerability and coping capacity to the overall damage" → I would add hazard among damage drivers

*R: done*

Pg. 1 line 29: "For example, the development, calibration and validation of flood damage models requires detailed, object-specific damage data as well as comprehensive information about exposure and vulnerability characteristics" → and hazard

*R: done*

**Section 2**

Pg. 3 line 85: "An example for a continent-wide database" →    I would say "continent-wide event database"
*R: done*

Pg.3 line: "In recent years, many initiatives were launched at national and European Union (EU) levels to improve the availability and usefulness of damage data. For example, an ongoing EU initiative aims at the standardization of damage databases. Based on a defined conceptual framework, this initiative provides technical recommendations for the development, of EU guidelines for recording disaster impacts with the central aim of translating the Sendai Framework for Disaster Risk Reduction into action (Corbane et al., 2015)" →    This sentence is not linked with previous and following ones. I suggest removing.
*R: done*

Pg. 4 line 11: "The main objective of this database is the development of specific depth-damage curves for Italian contexts. For this, new procedures for data collection and storage were developed and applied at the local level for the residential and commercial sectors (Molinari et al., 2014)"
→    As to my knowledge, this is not correct. The main objective of the database (i.e. FloodCAT) is to address knowledge needs required by the Floods Directive. There are not standardised procedures for data collection linked to the database, and the database works at the event level (not object level). Please, amend.
*R: we removed these sentences/this example from the text.*

**Section 3**
Pg. 5 line 148: "The data structure for HOWAS21 was derived from a multi-step online expert survey based on the Delphi-approach. The central idea of this approach is to reach a consensus among the respondents by having a questionnaire filled several times, after receiving feedback of earlier responses of all participants" →    I think an extended description of the adopted Delphi methodology is required for readers who are not familiar with previous works of the authors' team.
*R: we added some more information about the undertaken Delphi-approach: "To address the needs of different professional fields, 55 experts working in the field of flood damage analysis for government entities, the (re-)insurance industry, engineering consultancy and science were included in the survey panel (Elmer et al., 2010a).  The expert survey consisted of three rounds and was conducted online as a "Tele-Delphi". In the first round the panelists chose one or more of six sectors to answer, for which they were asked to evaluate the importance of a number of variables for flood damage analysis. New variables could be added. In the second round the median of the answers of round one was given as feedback.  Variables for which consensus about their importance had already been achieved were not reconsidered. Variables that were added by the experts during the first round could be evaluated by the whole panel in the second round. In the third round, experts were asked to rank by importance those variables for each sector that got the highest ratings in the first two rounds."*

Pg. 6 line 161: "The attributes of individual damage cases are grouped into three (partly sector-specific) database tables as shown exemplarily in Table 1" →    I understand Table 1 refers only to sector 1 & 2. If this is the case, please specify.
*R: we specified now, that Table 1 shows only one sector as example, namely private households.*

Pg. 6 line 176: "In total, HOWAS21 incorporates a broad range of hazard variables (e.g. flow velocity, flood duration, and contamination), vulnerability parameters such as building characteristics (e.g. building shape, year of construction) and precautionary measures (e.g. warning time, type and effectiveness of measures), and flood consequences (e.g. absolute and relative damage of flood-affected objects, economic damage due to business interruption in the commercial sector"
→    REPETITION, see pg. 6 line 162-163
*R: we moved the additional text on page 6 lines 176 ff to page 6 line 162-163, to avoid repetition.*

Pg. 6 line 188: "This interface is directly accessible using a standard web browser following the URL http://howas21.gfz-potsdam.de and can be used to visualize, analyze, and download HOWAS21 data" → authors should specify that the interface is only in German, this can be a big issue for the use of the database for international data
*R: As mentioned before, the database is now available in English and German, this is now stated in the text.*

Pg. 6 line 189: "Users are provided query functionality in the database on selectable criteria, such as catchments, damage sectors, or year of event" → REPETITION, see pg. 5 line 136
*R: Repetition is deleted*

Pg. 7 line 192: "A variety of attributes (e.g. river catchment, flood event year, damage sector) can be used to filter and analyze the data" → REPETITION, see previous comment
*R: Repetition is deleted*

Pg. 7 line 198: "It was developed and maintained by the German Working Group on water issues of the Federal States and the Federal Government (LAWA) from 1978 to 1994 (Buck and Merkel, 1999; Merz et al., 2004). Damage data of HOWAS were collected via on-site expert surveys by damage surveyors 200 of insurance companies and used as a basis for financial compensation, wherefore these damage estimates are considered to be reliable" → REPETITION, see section 2
*R: Repetition is deleted*

Pg. 7 line 213: "…in the City of Dresden affected during the Elbe river flood in 2002" → REPETITION, see the line above and line 219
*R: Repetition is deleted*

Pg. 7 line 216: "Second, the magnitude of structural damage was quantified on a six-point scale, and the condition of the road before the flood was quantified on a five-point scale by experts from the city administration (Kreibich et al., 2009)" → this is not clear, neither are the motivation of this process. Please, explain
*R: Since the information contained in HOWAS21 is focused on the monetary damage, we deleted this sentence to avoid confusion.*

**Section 4**
Pg. 8 line 221: "HOWAS21 aims at compiling comprehensive flood damage data (i.e. including object-specific hazard, exposure, and vulnerability characteristics) to support forensic flood damage analyses as well as flood damage model derivation" → REPETITION, said many times; please, consider removing
*R: Repetition is deleted*

Pg. 8 line 228: "Further, damage in the sectors public thoroughfare and water courses and hydraulic structures is by definition classified as building damage" → why? not clear; please, specify
*R: we provide some examples to clarify this: "Further, damage in the sectors public thoroughfare and water courses and hydraulic structures is by definition classified as building damage, since e.g. roads, rail tracks, embankments and flood masonry walls are all constructions."*

Pg. 8 line 234: "Both in respect to the number of damage records and the level of detail of information, i.e. the number of different hazard, exposure, and vulnerability variables, HOWAS21 is the most comprehensive flood damage database for empirical data worldwide" → Have authors evidence on this?
*R: we don't have a proof, thus we added "to the best of our knowledge" in the text.*

Pg. 8 line 246: "in the variable space for the private households sector" →  What does it mean? Please, specify

*R: We reformulated the sentence to make it clearer: "In turn, for certain other non-mandatory variables such as building type or building shape of objects in the private households the data availability is close to 100%."*

Pg. 8 line 247: "Generally, the availability of detailed flood damage data is often limited due to the facts that damage data collection in the aftermath of a flood is not mandatory, sufficient and properly trained personnel is mostly not available, and collection standards do not exist" →  this is partly true for HOWAS21 as data were all collected by trained personnel, also data from the original HOWAS. Isn't it?

*R: We added the sentences: "This is a challenge for HOWAS 21 which relies on the supply of flood damage data acquired via collection campaigns undertaken by the community after flood events. A compromise had to be found between the wish to incorporate comprehensive datasets into HOWAS21 and the limited availability of damage data." And moved this statement to section 3.1 Concept and structure, since this general statement fits better there than in the descriptive statistics section.*

Pg. 9 line 252: "Damage data in the sectors public thoroughfare and watercourses and hydraulic structures are yet only available for the City of Dresden during the 2002 Elbe flood" →  REPETITION, see comment above

*R: Repetition is deleted*

Pg. 9 line 263: "More specifically, such analyses are performed e.g. to quantify the relative contribution of damage drivers such as exposure, vulnerability and coping capacity to the overall damage" →  I would replace with "More specifically, such analyses are performed e.g. *to understand* the relative contribution of damage drivers such as *hazard*, exposure, vulnerability and coping capacity to the overall damage"; as most of forensic investigations are conducted in the social/economic domain, on the bases of qualitative data.

*R: done*

Pg. 9 line 265: "Other applications include the assessment of interdependencies of damage drivers, and the change of the correlation between a specific damage driver (e.g. water depth) and the resulting consequence (damage) over time (e.g. between different events in the same region)" →  I would not say these are typical applications in forensic investigation.

*R: Agreed, we deleted this sentence.*

Section 4.2.1 →  it is not clear to me what authors mean here with "interactions among damage-influencing variables" and how the performed analysis gives information on that; please, explain

*R: the sentence has been rephrased and now reads as follows: "In order to supplement the investigation of information value and redundancy with an estimation of the influence of the PCs on flood damage, the correlation between the factor scores of each PC and the absolute building damage was analysed."*

Pg. 11 line 341: "the following exemplary analysis is aimed at identifying potential trends in the linear relation between water depth and absolute building damage to private households between different flood years using multilevel regression" →  how such information can support decision-makers? Please, specify

*R: We added the following sentence: "For decision making in flood risk management it is important to know if the vulnerability of exposed objects (e.g. residential buildings) is changing over time; e.g. a decreasing trend in vulnerability might confirm the effectiveness of an implemented risk mitigation strategy; an increasing trend in vulnerability might indicate that risk management (and communication) needs to be improved."*

Pg. 12 line 360-374 → check figure numbers
*R: figure numbering is now corrected*

Pg. 12 line 363: "the high group variability of both regression parameters among the 13 flood years clearly confirms significant group effects as already suggested by the ICC" → not clear, please specify
*R: We added some additional explanation: "Looking at the lines reveals a more or less continuous increase in both the intercept and the slope with increasing flood event year. **Increasing intercept and slope point to increasing vulnerability, since the same water depth leads to higher damage.** This trend can be further investigated when plotting the group effects of the model (see Fig. 6). Three main findings emerge: First, the high group variability of both regression parameters among the 13 flood years clearly confirms **significant differences in vulnerability between the flood events**, as already suggested by the ICC."*

Pg. 13 line 394: "An exception is, however, the public thoroughfare sector, for which the polynomial regression curve is noticeably undulating" → Could authors explain why?
*R: This strange shape of the polynomial regression is unrealistic and due to strangely structured data, i.e. only very few different water depth values for which damage data is available. We added the following to the text: "An exception is, however, the public thoroughfare sector, for which the polynomial regression curve is noticeably undulating, **which is unrealistic and most likely due to a measurement artefact, since damage data is only available for very few water depth values, pointing to excessive aggregation.**"*

**Table and Figures**
Figure 5 → please check caption; I guess on the right slope is depicted
*R: this is correct and already stated in the figure caption: "The right plot depicts the random slopes."*

Figure 6 is not comprehensible: too little and too blurry
*R: Figure 6 (new Figure 7) has been improved for better readability.*

Table 1 → I would add a column for damage attributes
*R: According to the database structure, damage attributes are included in the second column.*

[revised manuscript text omitted]

---

## Author Comment (AC2) · 29 May 2020

**Authors response to Referee #2**

*Dear referee, we would like to thank you for the time and effort put into reviewing the manuscript. This response carefully addresses all the comments (our response is marked with an R: and written in italics). We further attach a change tracked version of the manuscript from which the changes proposed can be seen, see below this response.*
*Best regards*
*Heidi Kreibich on behalf of all co-authors*

**Referee #2**
The paper presents the flood damage database HOWAS21, and its use to support both forensic flood investigation and flood damage models. The topic is significant and has a great scientific interest. Nevertheless, I think that the paper should be organized in a more strict way, reducing the long descriptions, avoiding randomized examples and clearly dividing different features discussed using bullet points.
*R: Thank you for the generally positive feedback and for the helpful suggestions for improvement. We have improved the structure, included some sub-headings and reduced unnecessary repetitions, so that we hope that the text is now organized in a clearer way.*

The parts of the paper describing the db are not very clear and the link with floods that caused damage is missing. Moreover, I suggest to describe the procedure more clearly because it is not clear how other researchers could easily implement the same kind of analysis.
*R: We improved the description of the database, the recent enhancement to an international database as well as of the methods section to clarify how the analyses can be implemented. At the beginning of the section "4.1 General descriptive statistic" we added some information about flood events. However, we also clarified at the beginning of the section "3.3 Data sources" that HOWAS 21 is designed for empirical flood damage data which stem from various damage data acquisition campaigns, of which some, but not all, were undertaken after specific flood events. See also response to comment below.*

In the following the points that in my opinion could be improved:
Section 2 is long and sometimes confusing. It should be rearranged. I think that, in order to help the reader to follow the description, table S1 should be included in the paper (formatted in a more concise way, by clustering groups of similar db). I also suggest to introduce some bullet points or sub-headings, because currently this section is long and dispersive.
*R: We introduced sub-headings into section 2 to make it clearer and more structured. However, we would rather avoid the large Table S1 in the main text, since we believe that it would rather not improve the readability and conciseness of the manuscript.*

Section 3.1: I suggest to schematize this part in order to be more effective. Firstly, the type of users should be clearly divided and not described altogether (also a diagram could be useful).
*R: we included a new figure/diagram showing the different user groups*

At line 176: "In total, HOWAS21 incorporates a broad range of hazard variables" and after this the Authors presented examples randomized of these variables. I suggest to list in a table all the variables. This is a paper based on a db: it is impossible for the reader understand the discussion without a clear idea of the variables included. I understand that the db has been presented in previous papers, for this reason a simple and clear table can be easily prepared and can be more explicative

than a series of examples.

*R: Listing all variables for all sectors would lead to very long lists of variables. We believe that this would make the manuscript very hard to read, with disturbing long lists of variables included. We therefore list all variables of one sector, and include a link (DOI) to the recently updated web-site where all variables of all sectors can be found.*

Section 4.1. This section starts with damage described per type of damaged element without talk about the events that caused damage. How many flood events are included? What types are the most frequent? What the regions affected. This focus on the effects (damage) neglecting the causes (floods) also affect figure 1.

*R: At the beginning of the section "4.1 General descriptive statistic" we added the following information about flood events: "HOWAS21 comprises a total number of 8558 object-specific flood damage records from 14 years with flood events between 1978 and 2013 in Germany (see Fig. 2). **The majority of available damage cases was caused by river floods, and a small additional amount of cases can be attributed to groundwater flooding or pluvial flooding.** The geographical distribution of the damage data is depicted in Fig. 2, **most damage cases occurred in the Federal State of Saxony followed by the Federal State of Bavaria. Most damage cases resulted from the flood events in June 2013 in the Elbe, Danube, Rhine, and Weser catchments, in August 2002 in the Elbe and Danube catchments and the more localized event in June 1984 in the Tauber catchment."** Additionally, we made clear, that the database is not event specific, but that data integration is rather according to data acquisition campaigns. We added at the beginning of the section "3.3 Data sources" the following explanation: "The Flood Damage Database HOWAS21 is designed for empirical flood damage data which stem from various damage data acquisition campaigns. Some of these campaigns were undertaken specifically after large flood events, others were dedicated to specific flood types and collected all flood damage data available irrespective of specific flood events. Thus, not all data sets are associated to a specific flood event, e.g. damage due to small localized events and/or pluvial or groundwater flooding. It is not in scope of HOWAS21 to provide a flood event definition."*

Fig. 1 is not homogeneous. On the Y axis I see "flood 2013; Elbe 2006; GW flooding 2006". What is the criterion to name the events analyzed? The basin? I don't think, because it is not reported in all the records. The year? I don't think, because there is one record without the year. The type of flood? I don't think, because it is not specified in all the cases. This is an important point to homogenize because if the events are well known to the authors, they can be unknown for readers. It is also unclear what the x axis reports. Maybe the number of cases of damage? "Number of data records" is very vague. In the map: what "community" means? Village? Town? Municipality? Prefecture? It is unclear what the colored areas represent.

*R: We agree that the labeling in the figure was inconsistent and now provide an improved version that highlights the main flood type and the year of the flooding.*

Concerning the variables reported in fig 3, it is not clear how some of them are measured. For example to me it is not clear how "building shape" or "roof type" are represented as numerical variables.

*R: We included more details about the variables into Table 1 to clarify these aspects.*

Conclusions are not very representative of the contents of the paper, I suggest to review.

*R: We have included more information about the data analyses into the conclusions, to make them more representative for the whole article.*

Fig. 6 is absolutely not readable.

*R: Figure 6 (new Figure 7) has been improved for better readability.*

[revised manuscript text omitted]

---

## Referee Report (RR1)

Journal: NHESS
Title: **The object-specific flood damage database HOWAS21**
Author(s): Kellerman et al.
MS No.: NHESS-2019-420
MS Type: Research Article
**Iteration: Second review**

I really thank the authors for the efforts put in revising the manuscript. With respect to the two main criticalities I highlighted in my previous review, I think that the explanations of the technical and contents improvements with respect to the version described in Keibrich et al. (2017) overcome the problem of replicability. Still, I am still sceptical on the novelty of applications described in the second part of the paper, whose implementation are not linked to the features od the database itself (e.g., on the usability of information on data quality, on multi-sector data availability) but to the availability of data; indeed, discussed applications are not novel for the authors research group but simply refer to a larger dataset. I agree with authors that the presentation of several exemplary analyses of the HOWAS21 data provides motivation for the community to increase efforts for the collection of more such detailed object-specific damage data, undertake more such forensics analyses and develop better damage models beyond depth-damage functions. I also agree that it is the first time that an analysis of such a big data is performed, leading to some novel results in terms of damage dynamics and drivers in Germany. But, inferring new knowledge from the dataset is not the declared objective of the authors, which is instead i.e. "highlighting exemplary analyses to demonstrate the use of HOWAS 21". I leave the editor the final decision on this, being my and authors visions in contrast.

Beyond this aspect, the paper was already well written and organised and the new version of the introduction makes the paper even more clear and robust. From this perspective, the paper can be published as its, after the corrections of some typos.